# Observation of topological states residing at step edges of WTe$_2$

Lang Peng [1], Yuan Yuan[1], Gang Li[2,3], Xing Yang[1], Jing-Jing Xian[1], Chang-Jiang Yi[4], You-Guo Shi[4] & Ying-Shuang Fu[1]

Topological states emerge at the boundary of solids as a consequence of the nontrivial topology of the bulk. Recently, theory predicts a topological edge state on single layer transition metal dichalcogenides with 1T' structure. However, its existence still lacks experimental proof. Here, we report the direct observations of the topological states at the step edge of WTe$_2$ by spectroscopic-imaging scanning tunneling microscopy. A one-dimensional electronic state residing at the step edge of WTe$_2$ is observed, which exhibits remarkable robustness against edge imperfections. First principles calculations rigorously verify the edge state has a topological origin, and its topological nature is unaffected by the presence of the substrate. Our study supports the existence of topological edge states in 1T'-WTe$_2$, which may envision in-depth study of its topological physics and device applications.

[1] School of Physics and Wuhan National High Magnetic Field Center, Huazhong University of Science and Technology, Wuhan 430074, China. [2] School of Physical Science and Technology, ShanghaiTech University, Shanghai 200031, China. [3] Institute of Solid State Physics, Vienna University of Technology, A-1040 Vienna, Austria. [4] Institute of Physics, Chinese Academy of Sciences, Beijing 100084, China. Lang Peng and Yuan Yuan contributed equally to this work. Correspondence and requests for materials should be addressed to G.L. (email: ligang@shanghaitech.edu.cn) or to Y.-S.F. (email: yfu@hust.edu.cn)

In the two-dimensional (2D) topological insulators (TIs)[1, 2], the nontrivial edge state (ES) supports quantum spin Hall (QSH) effect, where the electrons at the edge of the system possess different spins when propagating along opposite directions. This enables the topological ES immune to scattering from non-magnetic impurities as protected by the time-reversal symmetry. Since the early proposal of QSH effect in graphene with spin-orbit coupling[3] (SOC), a number of 2D TI systems that are more readily examined by experiments are predicted[4]. However, only a few of them are confirmed experimentally, including quantum well structures of HgTe[5] and InAs[6], thin layers of Bi[7], Bismuthene[8] and FeSe[9], as well as the step edges of bulk $ZrTe_5$[10, 11], Bi[12], $Bi_{14}Rh_3I_9$[13], and (Pb,Sn)Se[14].

Recently, a family of 2D TI based on single-layer transition metal dichalcogenides (TMDs) with $1T'$ structure is predicted theoretically[15]. Combining the capability of van der Waals (vdW) stacking, the TMD QSH insulators provide the advantage of multiple edge conduction channels, which is highly desirable for practical device applications. This stimulates intensive research interests in identifying the topological phases in single layer TMDs, especially how to tune the crystal structure into the $1T'$ phase[16–18]. Recent experimental achievements have made it possible to grow $1T'$-$MoTe_2$ single layers by chemical vapor deposition[19] or molecular-beam epitaxy[20]. Moreover, ultra-thin[21] and even single layer[22] $1T'$-$WTe_2$ devices are successfully fabricated with mechanical exfoliation method. Tantalizing evidence of a positive QSH gap and edge conduction signatures are observed with transport measurements[21, 22]. Particularly, the edge conduction in monolayer $WTe_2$ show several features of topological character, such as suppression with in-plane magnetic field and absence in bilayer systems[22]. However, the measured conductance does not reach quantum conductance, leaving its assignment as topological origin not conclusive.

$WTe_2$ has the $1T'$ structure in its natural ground state and exhibits the largest SOC strength among TMDs. These properties render it the most promising candidate to search for the predicted QSH state[15]. Moreover, $WTe_2$ is predicted to possess nontrivial topological phases in its bulk as type-II Weyl fermions[23]. Step edges of the bulk $WTe_2$ offer the interfaces between the $WTe_2$ layer and the vacuum, which are distinct in electronic topology. In this regard, compared with the monolayer $WTe_2$ that is challenging to fabricate, the bulk step edges may be an appealing alternative to examine the potentially hosted topological ES.

For this, we study the electronic states of step edges of $WTe_2$ with spectroscopic imaging scanning tunneling microscopy, which is a powerful probe sensitive to the local density of state (LDOS) with high energy and high spatial resolution. We show the emergence of topological ES residing at step edges of $WTe_2$, which exhibits robustness against edge irregularities. Their topological nature is substantiated by theoretical

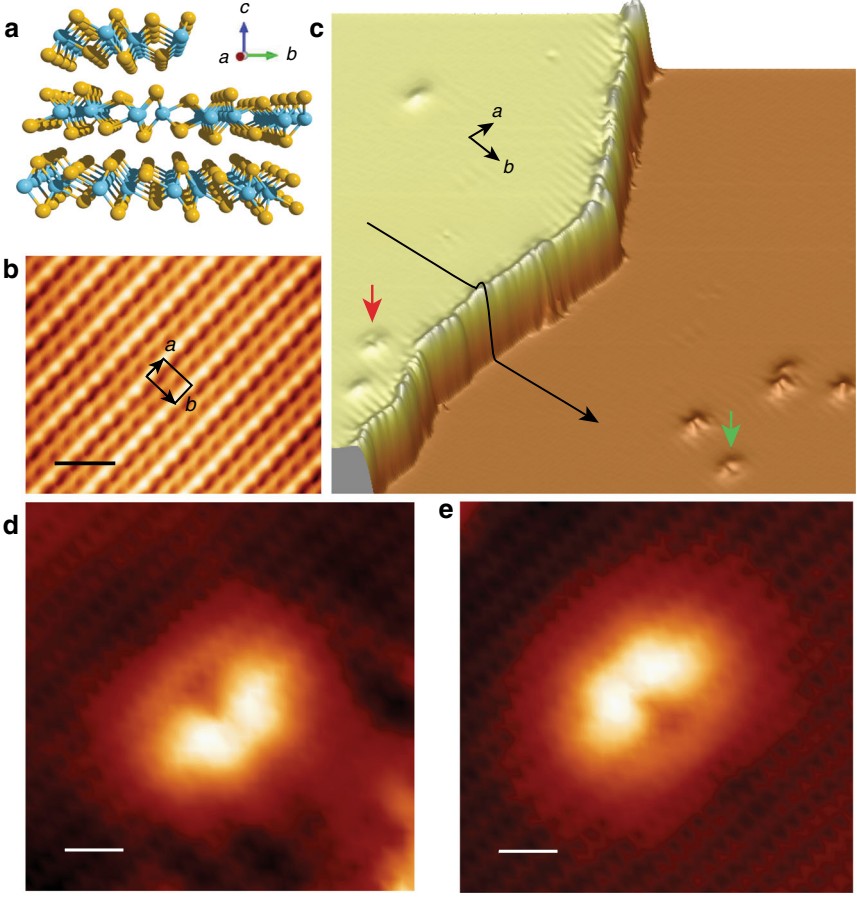

**Fig. 1** Topography of $WTe_2$. **a** Schematic illustrating the crystal structure of $WTe_2$ with a step edge. The W and Te atoms are colored with *cyan* and *orange*, respectively. **b** STM image showing the topography of $WTe_2$ with atomic resolution. Imaging conditions: $V_s = 120$ mV, $I_t = 100$ pA. The unit cell of the (001) plane is marked. **c** Pseudo 3D image of $WTe_2$ showing a step edge. Image size: 60 × 60 nm. The *black line* is a sectional line drawn across the step. The bright protrusions on both the upper and lower terraces are crystal defects. The *red arrow* and *green arrow* mark two typical defects that are mirror-symmetric to each other. Their zoom-in STM images are shown in **d**, **e**, respectively. The wave like patterns around the defects and steps in **c** are the electron standing waves. Imaging conditions of **c**, **d**, **e**: $V_s = 150$ mV, $I_t = 200$ pA. The *scale bar* in **b**, **d**, **e** corresponds to 1 nm

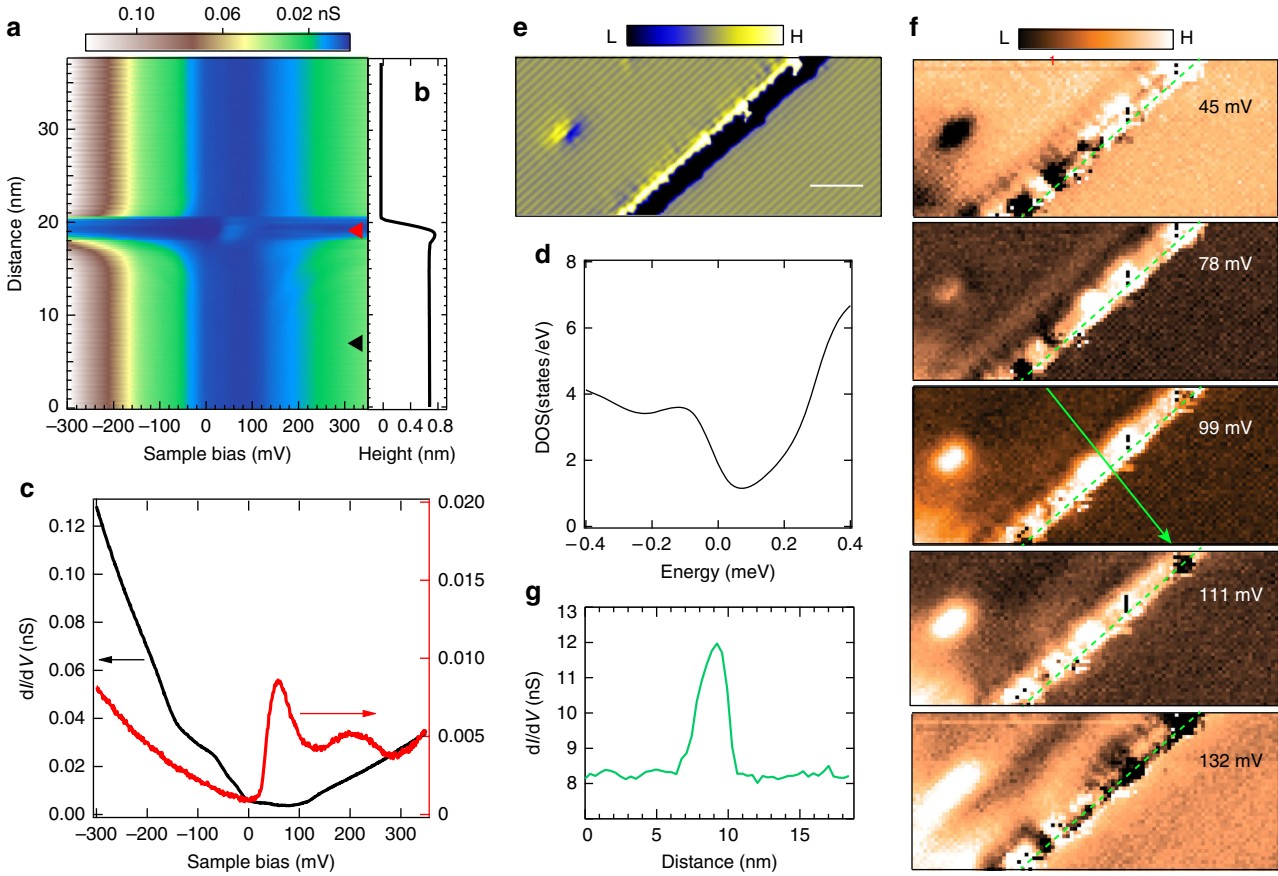

**Fig. 2** Spectroscopy of the edge states of WTe$_2$. **a** Two-dimensional conductance plot of tunneling spectra measured along the *black line* in Fig. 1c, whose sectional line is shown in **b**, showing the emergence of edge states around the step edge. **c** Typical tunneling spectra measured at the step edge (*red curve*) and at a location at the inner terrace (*black curve*). The spectra were extracted from **a** at locations marked with *red* and *black triangles*, respectively. Measurement conditions of **a**: $V_s = 150$ mV, $I_t = 200$ pA and $V_{mod} = 3.5$ mV$_{rms}$. **d** Calculated density of states of bulk WTe$_2$ with DFT. **e** Derivative STM image of a single layer high step edge along *a*-direction. The left terrace is the higher terrace. The *scale bar* corresponds to 5 nm. Measurement conditions: $V_s = 180$ mV, $I_t = 200$ pA. **f** Spectroscopic mapping of the imaged area in **e** at different voltages, showing the spatial distribution of the edge state with energy. Measurement conditions: $V_s = 180$ mV, $I_t = 600$ pA, $V_{mod} = 3.5$ mV$_{rms}$. **g** A sectional line profile extracted from **f** along the *green line*. The *green-dashed lines* in **f** mark the position of the step edge

calculations. Our study paves the way for incorporating bulk properties to the topological ES and developing topological device applications.

## Results

**Step edge and two surfaces of WTe$_2$.** WTe$_2$ has a layered structure with vdW bonding between the layers. Its $1T'$ structure originates from a lateral distortion of W atoms towards the $b$ direction of the $1T$ structure. This creates zigzag chains along the $a$ direction with neighboring chains having different heights (Fig. 1a and Supplementary Fig. 1a). The distortion directions of the adjacent WTe$_2$ layers are opposite. Thus, the top and bottom surfaces are of different type. After cleaving, both surfaces can sometimes co-exist that are separated with single-layer high step edges. Figure 1c shows a STM image of such surfaces with a step. Its zoomed-in image clearly resolves (Fig. 1b) the atomic resolution of the Te atoms of the WTe$_2$ (001) surface, which exhibits alternating bright- and dark-chain structures. Standing wave patterns in the vicinity of the step edges and defects seen in Fig. 1c propagate along the chain direction, reflecting the anisotropic character of the band structure in WTe$_2$. The step height is measured as ~ 0.72 nm, i.e. single layer high WTe$_2$, demonstrating the upper and lower terraces are two different

surfaces. Such assignment is augmented by the appearance of defects, which mark the type of surfaces via symmetry. Figure 1d, e show two identical kinds of defects at both surfaces. They point to opposite directions, clearly revealing the two surfaces are different.

**Spectra of ES.** To unravel its electronic structure, we acquire tunneling spectra (Fig. 2a), which is proportional to the LDOS, along a line perpendicularly across the step edge in the $a$-direction (Fig. 1c, *black line* and Fig. 2b). It is seen that the spectra at the inner terraces are spatially homogenous and are identical on both surfaces. Upon approaching the step edge, the tunneling conductance enhances drastically than that of the inner terrace between ~ 50 meV and ~ 130 meV. This indicates the existence of prominent ES. To inspect more details, we extract the spectra obtained at the step edge (Fig. 2c, *red line*) and at the inner terrace (Fig. 2c, *black line*). Evidently, the tunneling conductance of the inner terrace has finite value for all energies, consistent with WTe$_2$ being a semi-metal. Its spectroscopic shape is captured nicely with our calculated density of states with density functional theory (Fig. 2d and Supplementary Fig. 1). For the spectroscopy at the step edge, there appear two peaks at ~ 60 meV and 200 meV, respectively. The detailed shape of the ES

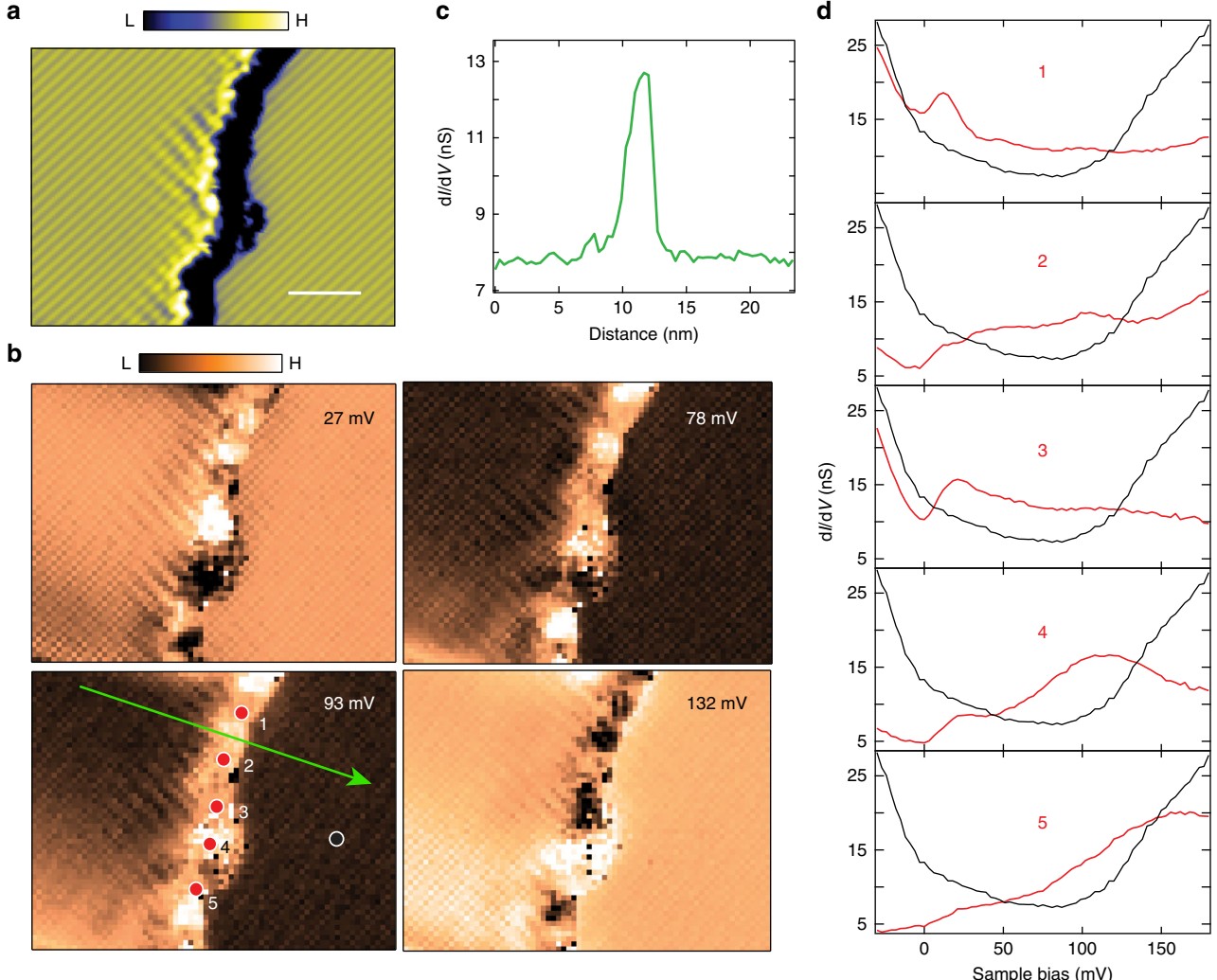

**Fig. 3** Edge states of $WTe_2$ residing at an irregular shaped step edge. **a** Derivative STM image of a single layer high step edge with irregular shape. The *scale bar* is 5 nm. Measurement conditions: $V_s = 180$ mV, $I_t = 200$ pA. **b** Spectroscopic mapping of the imaged area in **a** at different voltages, showing the spatial distribution of the edge state with energy. **c** A sectional line profile extracted from **b** along the *green line*. **d** Tunneling spectra (*red curves*) at different locations of the step edge (*red dots* in **b**). The spectroscopy (*black curve*) of the inner terrace (*black dot* in **b**) is shown for comparison. Measurement conditions of **b**, **d**: $V_s = 180$ mV, $I_t = 600$ pA, $V_{mod} = 3.5$ mV$_{rms}$

varies at different locations of the step edge (Supplementary Fig. 2). This implies the atomic geometry of the step edge is not uniform, as is also observed in the topological ES of many other 2D TI systems[9–11, 13]. During cleaving, step edges are formed by breaking the atomic bonds between the $WTe_2$ layers. The resulted edge geometry can exhibit complications such as different atom terminations, local relaxations and reconstructions, etc.

**Spatial distribution of topological ES**. Next, we did real space spectroscopic mapping to a straight step (Fig. 2e) along the *a*-direction to clarify the nature of the ES. Remarkably, the ES intensity enhances between ~ 50 meV and 120 meV, and gradually depresses outside the energy window. (Spectroscopic mappings at representative energies are shown in Fig. 2f). The ES is localized precisely along the step edge, revealing its origin and also confirming it is a 1D state. A conductance profile across the ES shows it has a lateral spatial extension of ~ 2.5 nm (Fig. 2g). This is of similar size as the topological ES observed in other systems[7–11].

More importantly, scrutiny on an irregular shaped step edge (Fig. 3a) indicates the robustness of the ES. Remarkably, the

conductance intensity of the ES exhibits the identical energy dependence (Fig. 3b) and the equal lateral spatial distribution (Fig. 3c) as the straight edge in Fig. 2e. It is noted that the spectroscopic shape of the ES varies at different locations of the irregular step edge (Fig. 3d), resulting in conductance fluctuations of the ES intensity. Nevertheless, the inspected locations entirely exhibit the ES prominently. We have also examined the spectroscopy of defect states and excluded the possibility of the observed ES is defect-induced (Supplementary Note 3).

**Discussion**
The trivial and nontrivial ES are from very different origins. The trivial ES comes from the termination of atomic lattices, such as that of $2H-MoS_2$[24]. Its spectroscopic feature is closely associated with the specific step direction, and can be scattered by defects. In contrast, the nontrivial ES originates from the nontrivial topology of the bulk and is irrelevant to the detailed shape of the step edges. The ES observed here not only shows robustness to the irregular shape of the step edges, but also exhibits the same energy dependence of conductance intensity and the equal lateral width along the different edge directions (Fig. 2, Fig. 3 and

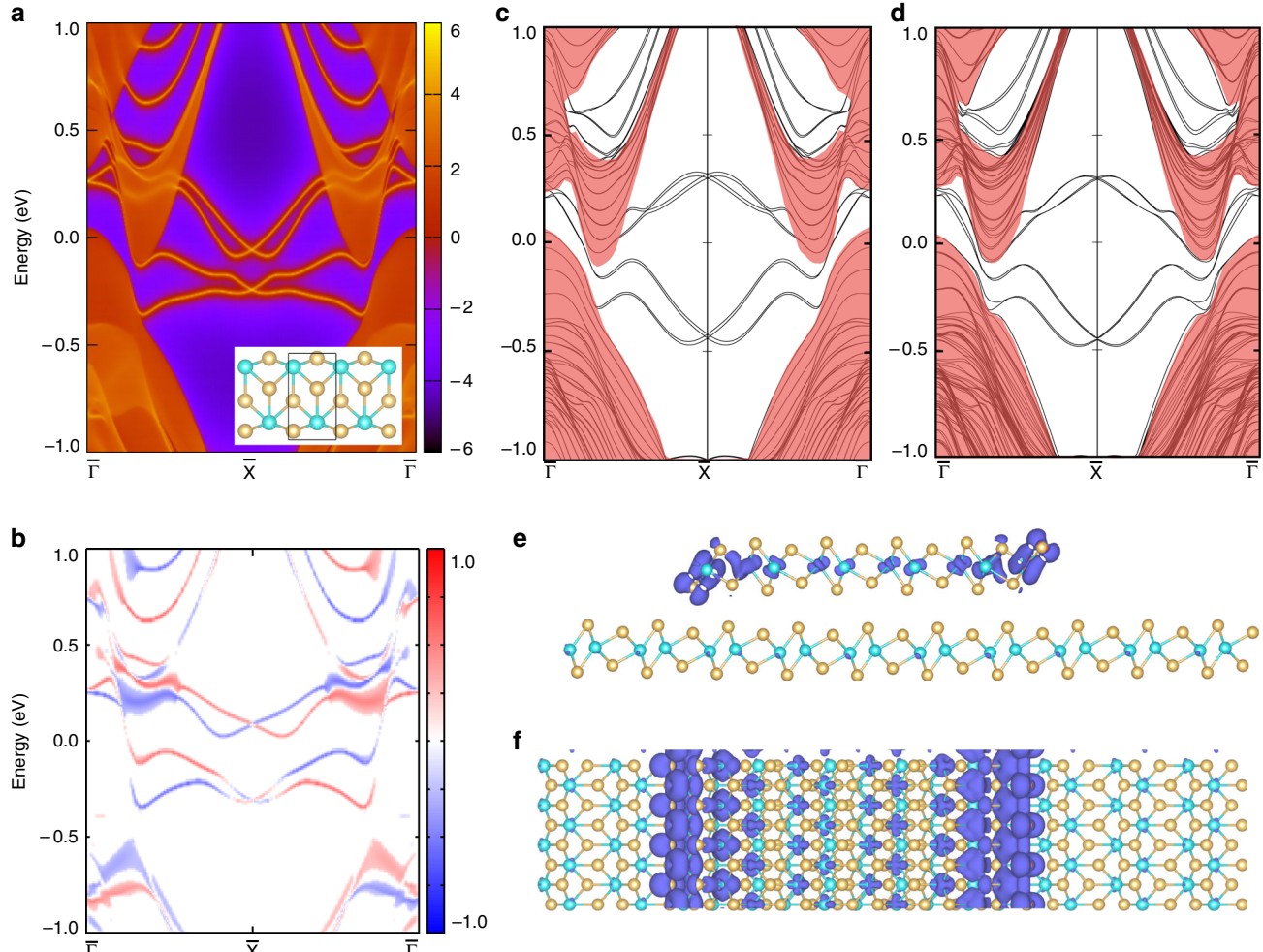

**Fig. 4** Calculated topological edge state of WTe$_2$. **a** The topological ES is calculated from the iterative Green's function approach based on a tight-binding model constructed from the monolayer 1$T$'-WTe$_2$ bulk electronic structure. **b** The spin polarization of the edge states, here the *red* and *blue color* characterize their opposite spin z-components. As also indicated in **a** the edge states in the two Dirac cones with same spin polarization connect to each other at $\overline{\Gamma}$. Thus, they are both topological edge states. **c** The electronic structure of monolayer 1$T$'-WTe$_2$ with fully relaxed edges calculated with ab initio method. **d** Same as **c**, but from a step edge with another bottom layer of 1$T$'-WTe$_2$ added. The bottom layer is periodic in both $a$ and $b$ directions. The region colored in *light-red* in **c**, **d** represents the states from the bulk, whereas the rest are the states staying at the edges. **e**, **f** The charge distribution of the ES in real-space from **e** the side view and **f** the top view

Supplementary Fig. 3). These phenomena can hardly be possible for trivial ES, but are much more likely related to the topological origin of the ES.

A more rigorous evidence of their topological nature is theoretical calculations, which have demonstrated fabulous success in identifying the topological phase of matter. To this end, we perform density-functional calculations. We start with a freestanding monolayer 1$T$'-WTe$_2$ ribbon by using the tight-binding model constructed from the bulk electronic structure. Its edges are terminated equivalently on both sides (Fig. 4a, inset), where no edge potential and structure relaxation are accounted. The ribbon width is chosen over the infinite limit to exclude hybridizations between the two edges. Both a $n$-field approach[25, 26] and a hybrid Wannier charge center[27] (Wilson loop[28]) method confirm the existence of topological ES (Fig. 4a, b and Supplementary Note 4). As a consequence of the nontrivial topology of the bulk, the existence of the topological ES should be irrelevant to the specific edge geometry. Indeed, our calculation confirms its existence at both $b$-edge (Supplementary Fig. 5) and differently terminated $a$-edges (Supplementary Fig. 6) without coexisting trivial ESs. Therefore, this supports the topological

nature of the experimentally observed ESs. It is found that the detailed spectroscopic features of the ES alter with different edge terminations (Supplementary Note 5), recalling the experimental observations. (Supplementary Fig. 2).

We then take the edge potential and structure relaxation into account, and calculate a 1$T$'-WTe$_2$ monolayer ribbon of 62 Å wide with the same edge termination as in Fig. 4a directly with ab initio calculation. We fix the shape of the slab and allow the internal coordinates of every atom to relax. Then, the bulk electronic structure and the ESs are calculated for this relaxed structure. As seen in Fig. 4c, the topological ES are clearly present with an increased energy separation compared with Fig. 4a. This calculated result is qualitatively consistent with ref. [15], despite the details of the band features are different owing to the different edge geometry and exchange potential.

To further evaluate the substrate effect, we consider a monolayer 1$T$'-WTe$_2$ ribbon of 30 Å width placed on another layer of it with ab initio calculation. After full relaxation, we observe small displacements of the atoms along the edge and a further separation of the edge layer from the substrate. Despite of the presence of the substrate and the induced geometrical relaxation,

the ES on the step edge shown in Fig. 4d not only qualitatively but also quantitatively resembles that of the freestanding monolayer edge (Fig. 4c). Furthermore, the charge distribution of the ES locates on the edge of the upper terrace, which substantiates their origin from the edge (Fig. 4e, f) and agrees with the experimental findings in Fig. 2f. This indicates the interlayer coupling does not alter the topological property of the ES owing to the weak coupling nature of the vdW stacking. This may have implications to other layer-stacked topological materials, which promises an abundance of unexplored experimental possibilities. Moreover, bulk WTe$_2$ is predicted to be a type-II Weyl semimetal, which features four small sections of disconnected topological surface Fermi arcs at its (001) surface[23]. The topological surface Fermi arcs are distinct from the topological ES at the step edge in momentum space, preserving the 1D nature of the topological ES[11].

Our study thus not only provides an experimental proof to the topological ES in monolayer WTe$_2$ albeit indirectly, but also expands the accessible scope of such nontrivial states to bulk samples. This is of particular importance for situations where bulk properties are desirably introduced into the system. For instance, bulk WTe$_2$ show superconducting phase transition under pressure[29, 30]. We have examined theoretically that the topological ES is preserved in the pressure regime where superconductivity of bulk WTe$_2$ coexists (Supplementary Note 6). This renders the step edge of WTe$_2$ a promising system for realizing topological superconductivity[9]. We envision that multiple conduction channels can be created for device applications by forming step edge arrays through the technologically compatible lithographic patterning technique, which is however hardly feasible for the monolayer, or further growth of nanostripes on WTe$_2$ substrate.

After completion of this manuscript, we became aware of two related works[31, 32]. Both works report the observation of topological ESs of single layer 1T'-WTe$_2$ grown with molecular beam epitaxy. The conductance of the observed topological ESs also indicates obvious variations as our studies.

## Methods

**Sample preparations and STM experiments.** The experiments were performed with a custom-made Unisoku STM (1300) at 4.4 K[33]. WTe$_2$ crystals grown by a solid-state reaction were cleaved in situ under ultrahigh vacuum conditions at ~ 77 K. After cleaving, the crystals were transferred quickly to the low temperature STM for subsequent measurements. An electro-chemically etched W wire was used as the STM tip. Prior to measurements, the tip was characterized on a Ag(111) crystal that had been cleaned by several cycles of Ar ion sputtering and annealing. The tunneling spectra were obtained by lock-in detection of the tunneling current with a modulation voltage at 983 Hz feeding into the sample bias.

**First principles calculations.** The WTe$_2$ monolayer was calculated with density functional theory. The projector-augmented-wave[34, 35] method implemented in the Vienna Ab Initio Simulation Package[36, 37] was employed with an energy cutoff of 500 eV. The generalized gradient approximation potential[38] was used in all calculations. The topological ES was calculated by applying the iterative Green's function approach[39] based on the maximally localized Wannier functions[40] obtained through the VASP2WANNIER90[41] interfaces in a non-self-consistent calculation with $9 \times 9 \times 1$ k-mesh.

**Data availability.** The data that support the findings of this study are available from the corresponding authors on reasonable request.

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

## Acknowledgements

The authors thank G. Xu, T. Hanaguri, S.-Q. Shen, Z.-W. Zhu, and X. Liu for helpful discussions. This work is funded by the National Key Research and Development Program of China (Grant No. 2017YFA0403501, 2016YFA0401003, 2016YFA0300604), the National Science Foundation of China (Grants No. 11474112, No. 11522431, No. 11474330). G. Li acknowledges the starting grant of ShanghaiTech University.

## Author contributions

L.P. and Y.Y. carried out the experiments with the help of X.Y. and J.-J.X. G.L. did the theoretical calculations. Y.-G.S. and C.-J.Y. grew the WTe₂ single crystal. Y.-S.F. supervised the project. Y.-S.F. and G.L. designed the experiment, analyzed the data and wrote the manuscript.

## Additional information

**Competing interests:** The authors declare no competing financial interests.

