## [Peer Review File · Nature Communications]

Reviewers' comments:

Reviewer #1 (Remarks to the Author):

This is a timely and important work on the postulated topological conductive edge state of 1T' WTe₂. The original prediction was on monolayer and stacked monolayers of same distortion, the authors showed the results also work for bulk terrace edges, with the top layer clearly in the 1T' structure from STM imaging and alternating Peierls distortion in the layer beneath. The conductive edge was shown to be robust regardless of disorder. I recommend publication after small revisions.

The authors should address the following questions.

(a) 1T structure can also have metallic edges, as shown in PHYSICAL REVIEW LETTERS 87, 196803. The authors should cite that work, and comment on the differences. For example, the authors may comment on whether 1T metallic edges are also robust against defects, with calculations.

(b) With calculations, the authors may comment on whether there is difference if the perturbation displacements are constrained to be mainly 2D, or mainly 3D (where the sheet is allowed to bend, and atoms are allowed to move quite far out of the plane)

Reviewer #2 (Remarks to the Author):

The manuscript describes STS data of a cleaved WTe₂ bulk crystal which is in the 1T' structure and presents compelling evidence for an edge state in an area of reduced dI/dV conductivity on the terraces away from the edge. Favorable comparison with DFT calculations implies that the edge state is of topological origin being located within an area of the band structure without projected bulk states. Albeit the material is not insulating, this result reveals additional evidence for the topologically insulating character of monolayer WTe₂ in terms of topology. This is a timely result that could have a large impact due to the van-der-Waals character of WTe₂ allowing for LEGO-stacking. Thus, the perspective of such a topological 2D material goes beyond the topological edge states that have been found on other systems by STS recently allowing a more straightforward exploitation of its transport properties.

The results are well presented in the manuscript with two exceptions, which should be improved prior to publication.

a) The introduction and the corresponding embedding of the result into the literature is rather sketchy. In particular the phrase "Despite that a few 1T'-TMD single layers and devices have been successfully synthesized or fabricated, conclusive remarks on the topological ES have still not been reached." is a bit misleading. In particular, [19] shows very nice transport results for exfoliated monolayers, which exhibit a conducting edge and an insulating bulk with the edge conductance being absent in the bilayer. The conductance is not e^2/h , but larger, i.e. the result is indeed not conclusive, but this discrepancy has not been improved by the present paper. Thus, a more detailed introduction on the status of the topology in WTe₂, maybe also including [28], [29] in more detailed is mandatory in order to interpret the importance of the current results correctly. Also the list of 1D edge states probed by STS is incomplete lacking arXiv:1608.00812, which shows an edge state of Bismuthene at the Fermi level within a huge band gap and Science 354, 1269 (2016) showing topological edge states protected by a mirror symmetry which have to be added.

b) The description of the topological analysis in S3 appears appropriate to an experimentalist in case without band gap, but it is not very clearly described. Firstly. It would be nice to know, how the separation of the BZ into rectangles has been made in detail. I guess that one selects a grid,

where a band gap can be defined in each rectangle, but that might be nice to be mentioned explicitly. More importantly the Wilson loop in case of many valence bands is not completely clear to me. Does Fig. S3b only apply to the upper "valence band" and, if yes, how do the authors make sure that lower bands are not topological as well.

I believe that both problems can be solved such that a publication in Nature Communications can be recommended afterwards.

Reviewer #3 (Remarks to the Author):

The manuscript "Observation of topological states residing at step edges of WTe₂" by Peng and co-workers describes STM and DFT results obtained on cleaved bulk samples. The authors observe features in their spectroscopy data which they interpret as an edge states. Based on a comparison with DFT calculations it is suggested that this edge state is of topological origin.

To my opinion the manuscript in its current state is not of sufficient clarity to justify publication in Nature Communications or any other scientific journal. In fact, I have serious doubts whether the claims made are correct at all. In the following I will detail my main points of criticism:

1. I find the usage of the term "quantum spin Hall (QSH) edge state" highly misleading. The QSH effect is measured in real transport measurements (not STM). Its occurrence is commonly explained by the existence of topological spin-momentum locked boundary states. Obviously, the manuscript by Peng and co-workers under discussion here does not present real transport measurements, so the authors should refrain from using the term QSH in context with their measurements and calculations.
2. On page 4, lines 77-79, and the supplemental material (Fig. S2) the authors describe a strong variation of the spectra observed at WTe₂ step edges. This is rather unusual and raises concerns whether the features observed are really related to edge states. Can the authors exclude that the density of states visible close to the position of the Weyl point are defect induced? For comparison, how do the spectra of the two defects presented in Fig. 1d,e look like?
3. Somewhat related to my concerns raised in point 2: Looking at the spectroscopic mapping data presented in Figs. 1e and 3b it becomes apparent that the observed patterns at step edges strongly vary with bias voltage. This is in stark contrast with the results of Ref. 11 where the absence of bias dependent variations was taken as a hint for forbidden backscattering, a direct consequence of the spin-momentum locking of topological boundary states. How do the authors explain the stark variation of features close to the Weyl point and how can they safely conclude that the peaks observed are not defect induced?
4. On page 5, lines 92-94, the authors claim that the presence of the electronic feature at A and B steps "demonstrates the emergence of the ES is not specific to a particular step direction, but is related to the nontrivial topology of the bulk." While I agree with the first part of the sentence (with the above mentioned concerns still being valid), I strongly disagree with the second part. I don't see how the fact that the feature is present at A and B steps is necessarily related to a nontrivial topology of the bulk.
5. Page 5, line 98-99: The authors write that "the ribbon width is infinitely large to exclude hybridizations between the two edges." What differentiates an infinitely wide ribbon from an infinite plane/surface? Is it still a ribbon at all?
6. Related to the DFT calculations: Why do the authors restrict their calculations to monolayer WTe₂ or supported monolayers? Wouldn't it be more realistic to calculate step edges under

periodic boundary conditions?

7. Fig. 4: How does the calculated electronic structure of the step edges presented in Fig. 4 correspond to the measured spectra? Can the authors present a DOS of the step edge similar to Fig. 2c for the bulk? More specifically: Can they reproduce the two peaks at about +50meV and 200 meV?

—

Some minor points:

A. On page 3, line 53n the authors describe the surface structure expected from the schematic representation of Fig. 1a as ``zigzag chains''. I only see straight lines in the schematic drawing of Fig. 1a as well as in the STM image presented in Fig. 1b. To my opinion there is no justification to use the term ``zigzag'' here.

B. Page 4, line 74: ``consisting'' must be replaced by ``consistent''

C. The authors one recent paper on one-dimensional topological edge states, Science 354, 1269 (2016). This papers shall also be cited along with Refs. 7-11.

D. Page 5, line 98: Replace ``inert'' by ``inset''

Reviewers' comments:

Reviewer #1 (Remarks to the Author):

This is a timely and important work on the postulated topological conductive edge state of 1T' WTe₂. The original prediction was on monolayer and stacked monolayers of same distortion, the authors showed the results also work for bulk terrace edges, with the top layer clearly in the 1T' structure from STM imaging and alternating Peierls distortion in the layer beneath. The conductive edge was shown to be robust regardless of disorder. I recommend publication after small revisions.

Thank the reviewer for recommending the publication of our work with small revisions.

The authors should address the following questions.

(a) 1T structure can also have metallic edges, as shown in PHYSICAL REVIEW LETTERS 87, 196803. The authors should cite that work, and comment on the differences. For example, the authors may comment on whether 1T metallic edges are also robust against defects, with calculations.

Thanks for the nice suggestion. We have included the paper to the reference list. Monolayer and bilayer MoS₂ are stable in 2H and 1T' structure. The 2H structure is a large gap semiconductor, and the 1T' structure is a semimetal. The 1T structure is unstable and is a semimetal according to our calculations shown in the figure below.

(a) Calculated bands of monolayer 1T -MoS₂. (b) Calculated bands of monolayer 2H -MoS₂.

In the mentioned work, monolayer MoS₂ is of 2H structure and has a calculated direct band gap of 1.64 eV. Its edge states are essentially from the substantial modification of the band structure of MoS₂ by the edges, and not from dangling bonds. As is seen from our calculated bands in the above figure, 2H-MoS₂ is a semiconductor, which is consistent with the calculations performed in the mentioned work. It does not possess an inverted band gap. Thus, its edge state is topologically trivial, and can be scattered by defects. This is in contrast to the case of 1T'-WTe₂, whose edge mode is topologically nontrivial. Additionally, monolayer WTe₂ crystallizes in 1T'-structure, instead of 1T or 2H. It was shown [G. Eda et al., ACS Nano 6, 7311–7317 (2012)] that the 1T structure of monolayer transition metal dichalcogenide is unstable in free-standing condition and undergoes a spontaneous transition to the 1T'-structure, which is what we studied in our work.

We have added a new paragraph starting from “The trivial and nontrivial ES are from very different origins...” to last part of Page 5 in the main text to clarify that issue.

(b) With calculations, the authors may comment on whether there is difference if the perturbation displacements are constrained to be mainly 2D, or mainly 3D (where the sheet is allowed to bend, and atoms are allowed to move quite far out of the plane)

In our calculations, the edge relaxation of the 1T'-WTe₂ stripe is allowed in all three directions. The optimized structure however demonstrates that the structural relaxations are mainly confined in the direction orthogonal to the edge. As is shown in the figure below, we set the edge along *a*-direction. The notable relaxations are mainly in *b*- and *c*-directions for the outmost edge atoms. Their average relaxation change is less than 0.2 angstrom. Since there is only one substrate layer in our calculation, the change of interlayer distance is relatively large. After relaxation, the substrate layer and the step layer are further separated apart by around 0.8 angstrom.

To make that point clearer, we have added “We fix the shape of the slab...” and “After fully relaxing the internal coordinates...” to the first and second paragraph of Page 7, respectively.

Reviewer #2 (Remarks to the Author):

The manuscript describes STS data of a cleaved WTe₂ bulk crystal which is in the 1T' structure and presents compelling evidence for an edge state in an area of reduced dI/dV conductivity on the terraces away from the edge. Favorable comparison with DFT calculations implies that the edge state is of topological origin being located within an area of the band structure without projected bulk states. Albeit the material is not insulating, this result reveals additional evidence for the topologically insulating character of monolayer WTe₂ in terms of topology. This is a timely result that could have a large impact due to the van-der-Waals character of WTe₂ allowing for LEGO-stacking. Thus, the perspective of such a topological 2D material goes beyond the topological edge states that have been found on other systems by STS recently allowing a more straightforward exploitation of its transport properties.

The results are well presented in the manuscript with two exceptions, which should be improved prior to publication.

Thank the reviewer for the positive comments of our work.

a) The introduction and the corresponding embedding of the result into the literature is rather sketchy. In particular the phrase “Despite that a few 1T'-TMD single layers and devices have been successfully synthesized or fabricated, conclusive remarks on the topological ES have still not been reached.” is a bit misleading. In particular, [19] shows very nice transport results for exfoliated monolayers, which exhibit a conducting edge and an insulating bulk with the edge conductance being absent in the bilayer. The conductance is not e^2/h , but larger, i.e. the result is indeed not conclusive, but this discrepancy has not been improved by the present paper. Thus, a more detailed introduction on the status of the topology in WTe₂, maybe also including [28], [29] in more detailed is mandatory in order to interpret the importance of the current results correctly.

Also the list of 1D edge states probed by STS is incomplete lacking arXiv:1608.00812, which shows an edge state of Bismuthene at the Fermi level within a huge band gap and Science 354, 1269 (2016) showing topological edge states protected by a mirror symmetry which have to be added.

Thank the reviewer for the nice suggestions. We have added more references and further detailed contents of “including quantum well structures of ...” and “Recent experimental achievements...” to the introduction in Page 2.

Since our manuscript appears on arXiv at the same time as Refs. 31 and 32 (Original Refs. 28 and 29), we would like to give more introductions on those works in the third paragraph of Page 8, and modify as “After completion of this manuscript, we became aware of two related works^{31,32}. Both works report the observation of topological ESs of single layer 1T'-WTe₂ grown with molecular beam epitaxy. The conductance of the observed topological ESs also indicates obvious variations as our studies. Interestingly, the 2D bulk interior exhibits a positive QSH gap.”

b) The description of the topological analysis in S3 appears appropriate to an experimentalist in case without band gap, but it is not very clearly described. Firstly. It would be nice to know, how the separation of the BZ into rectangles has been made in detail. I guess that one selects a grid, where a band gap can be defined in each rectangle, but that might be nice to be mentioned explicitly. More importantly the Wilson loop in case of many valence bands is not completely clear to me. Does Fig. S3b only apply to the upper “valence band” and, if yes, how do the authors make sure that lower bands are not topological as well.

I believe that both problems can be solved such that a publication in nature Communications can be recommended afterwards.

Thank the reviewer for suggesting us to present more detail calculations of the topological invariance.

We numerically evaluated it based on the scheme proposed by L. Fu and C. L. Kane [Phys. Rev. B 74, 195312(2006)]. As is correctly pointed out by the reviewer, this semimetal system does not have a band gap. It is thus not easy to define a Fermi level that clearly separates the valence bands from the conduction ones. As for the insulating systems, their topology is characterized by the integration of the berry curvature for all occupied bands. For the current system, we found that a momentum-dependent Fermi energy can still be used to effectively define the valence bands. It is easier in this case to employ the band index to label the valence. We calculated the berry curvature for bands with indices smaller than this number (determined in the calculations by the number of valence electrons). The topological obstruction method allows an efficient evaluation of the topological invariant in the discretized Brillouin Zone with $k_i = b_i/N_i * i$ for $i = 0, \dots, N_i-1$, where b_i is the reciprocal vector and N_i is the number of slices along b_i . As b_1 and b_2 are not necessarily orthogonal, the resulting plaquette does not have to be a square. Along the closed edges of each plaquette, we calculated the phase change of the wave functions. The total phase change in half of the Brillouin Zone yields the Z_2 topological invariant. If the total phase change is odd times of π , the system is topologically nontrivial. Otherwise, it is topologically trivial.

We have added “These plaquettes are obtained by ...” to the last paragraph of Page 5 in supplementary information to more clearly explain our methodology of topological analysis.

Reviewer #3 (Remarks to the Author):

The manuscript “Observation of topological states residing at step edges of WTe₂” by Peng and co-workers describes STM and DFT results obtained on cleaved bulk samples. The authors observe features in their spectroscopy data which they interpret as an edge states. Based on a comparison with DFT calculations it is suggested that this edge state is of topological origin.

To my opinion the manuscript in its current state is not of sufficient clarity to justify publication

in Nature Communications or any other scientific journal. In fact, I have serious doubts whether the claims made are correct at all. In the following I will detail my main points of criticism:

We thank the reviewer for carefully reading our manuscript. Our point-to-point responses to the reviewer's concerns are the following.

1. I find the usage of the term "quantum spin Hall (QSH) edge state" highly misleading. The QSH effect is measured in real transport measurements (not STM). Its occurrence is commonly explained by the existence of topological spin-momentum locked boundary states. Obviously, the manuscript by Peng and co-workers under discussion here does not present real transport measurements, so the authors should refrain from using the term QSH in context with their measurements and calculations.

Thank the reviewer for raising this point. The use of the term "quantum spin Hall edge state" is mainly inherited from the theoretical investigations, as commonly appeared in studies of 2D topological systems with time-reversal symmetry. The quantum spin Hall insulator is also named as 2D topological insulator. The corresponding quantum spin Hall edge state is also called the topological edge state. These terminologies are widely used in the studies of topological systems. For example, Ref. 9 reports the existence of quantum spin Hall edge state in FeSe/STO system with ARPES and STM measurements, instead of transport means. However, in order to respect the original meaning of the quantum spin Hall effect, we agree with the reviewer that our work is not a transport measurement and have changed the relevant terminology of QSH edge state to topological edge state.

2. On page 4, lines 77-79, and the supplemental material (Fig. S2) the authors describe a strong variation of the spectra observed at WTe₂ step edges. This is rather unusual and raises concerns whether the features observed are really related to edge states. Can the authors exclude that the density of states visible close to the position of the Weyl point are defect induced? For comparison, how do the spectra of the two defects presented in Fig. 1d,e look like?

According to Ref. 23, WTe₂ is theoretically predicted to be a type-2 Weyl fermions, whose Weyl points are located at around 52meV and 58meV. To date, several APRES measurements have been performed aiming to observe the topological Fermi arc states, which is an experimental proof of its bulk states as Weyl fermions. However, the existence of topological Fermi arc states is still in largely debate. Since type-2 Weyl fermions have a tilted Dirac cone, its electron density at Weyl points is not small. Moreover, the Weyl points are inside the bulk states. Those factors make the probe of Weyl points with STS technique difficult. We would like to clarify that our spectroscopic data presented in the manuscript do not show the evidence and the energy of Weyl points.

For the comparison of defect states and edge states, we have added additional data on the defect states and their spectroscopic mapping as Fig. S3 to the supplementary information.

3. Somewhat related to my concerns raised in point 2: Looking at the spectroscopic mapping data presented in Figs. 1e and 3b it becomes apparent that the observed patterns at step edges strongly vary with bias voltage. This is in stark contrast with the results of Ref. 11 where the absence of bias dependent variations was taken as a hint for forbidden backscattering, a direct consequence of the spin-momentum locking of topological boundary states. How do the authors explain the stark variation of features close to the Weyl point and how can they safely conclude that the peaks observed are not defect induced?

It's indeed that the conductance intensity of our edge states has strong variations. We believe the variation is related to the non-uniform atomic geometry of the step edge. During cleaving, step edges are formed by breaking the atomic bonds between the WTe_2 layer. The resulted edge geometry can exhibit complications such as different atom terminations, local relaxations and reconstructions, etc. As is seen from our theoretical calculations, different edge geometries can induce drastic changes in the spectroscopic features of the edge states (please see the comparison of Fig. 4a and Fig. S6). However, the topological nature of the edge state is unchanged, because that is guaranteed by the nontrivial topology of the bulk interior.

We stress that the strong conductance variation of the topological edge state is also observed in other 2D topological insulator systems. For instance, two groups report different spectroscopic feature of the topological edge state of $ZrTe_5$ (Ref. 10 and 11). The topological edge conductance intensity show strong variation in $ZrTe_5$ (Fig. S5 of Ref. 10) and $FeSe/STO$ (Fig. S16 of Ref. 9). Moreover, the topological edge state in Ref. 13 (original Ref. 11) is not uniform either. Its edge conductance in Fig. 3b and c show obvious variations. We have added "This implies the atomic geometry ..." to the last paragraph of Page 4 in main text to clarify that issue.

We didn't seek the forbidden back-scattering feature of the topological edge state due to the following two reasons. For one thing, the edge geometry of our step edge is not uniform, which excludes the basis of examining the back-scattering by local potential perturbations. For another, even if the edge state is topological, back-scattering still occurs if the topological edge state has an energy dispersion possessing even numbers of crossings at certain energies, such as the topological edge state in Ref. 12.

To exclude the observed edge states are defect-induced, we give the following arguments. First, there are very few defects in our crystal, as is seen from the topography of STM image in Fig. 1. Second, it is unlikely for the edge to be all defects. Third, most defects are featureless in spectroscopy. Fourth, the conductance intensity of step edge and defects show different dependence on bias. We have added additional data as Fig.S3 to the supplementary information to support our arguments, and a sentence "We have also examined the spectroscopy of defect states..." to the third paragraph of Page 5 in the main text.

4. On page 5, lines 92-94, the authors claim that the presence of the electronic feature at A and B steps "demonstrates the emergence of the ES is not specific to a particular step direction, but

is related to the nontrivial topology of the bulk.” While I agree with the first part of the sentence (with the above mentioned concerns still being valid), I strongly disagree with the second part. I don’t see how the fact that the feature is present at A and B steps is necessarily related to a nontrivial topology of the bulk.

Thank the reviewer for raising this important point. We mentioned in the manuscript that the edge state is present not only along straight step but also along irregular shaped step edge. The trivial and nontrivial edge states are from very different origins. The trivial step edges come from the termination of atomic lattices, whose spectroscopic features are closely related to the specific step direction. While the nontrivial edge states originate from the nontrivial topology of the bulk and are irrelevant to the detailed shape of the step edges.

The edge states observed here show three key features. First, they are robust against the irregular shape of the step edges. Second, they exhibit the same energy dependence of conductance intensity along the different edge directions (Fig. 2, Fig.3 and Fig. S3). Third, they have equal lateral spatial width (*i.e.* equal decay length) along the different edge directions (Fig. 2, Fig.3 and Fig. S3). These phenomena can hardly be possible for trivial edge states, but are rather much likely related to the topological origin of the edge state.

A more rigorous evidence of their topological nature is theoretical calculations, which have demonstrated fabulous success in identifying the topological phase of matter. By using both *ab-initio* calculations, we prove nontrivial band topology of the bulk interior. More importantly, the edge states along both *a* and *b* directions are all topological with no coexisting trivial edge states. Therefore, this supports the topological nature of the experimentally observed edge states.

We have clarified that issue in the newly added last paragraph of Page 5 and the modified second paragraph of Page 6.

5. Page 5, line 98-99: The authors write that “the ribbon width is infinitely large to exclude hybridizations between the two edges.” What differentiates an infinitely wide ribbon from an infinite plane/surface? Is it still a ribbon at all?

We apologize for causing the misunderstanding. In the calculation of edge states, we constructed a slab (both directly in DFT and in Wannier tight-binding model) with periodic boundary condition in one direction and an open boundary condition in the other direction. The ES will appear at the two open boundaries. Such a slab is usually still called as a ribbon. The spacing between the two edges is defined as the width of the ribbon. Strictly speaking, we have a cylinder type slab with two explicit edges. By increasing the width of the cylinder and meanwhile monitoring the coupling between the two edges, one can effectively reach the infinite width limit when the coupling of the two edges becomes negligibly small.

We have changed the phrase “infinitely large” to “chosen over the infinite limit” in the last paragraph of Page 6 to avoid the misunderstanding.

6. Related to the DFT calculations: Why do the authors restrict their calculations to monolayer WTe₂ or supported monolayers? Wouldn't it be more realistic to calculate step edges under periodic boundary conditions?

We have calculated, as shown in Fig. 4a, 4c and 4d, both the monolayer WTe₂ and the step edge (or the supported monolayers as called by the reviewer). Fig. 4c and 4d show their corresponding electronic structures. Their nice agreement clearly demonstrates that the presence of the supporting layer does not modify the electronic structures of the bulk and the edge essentially. Thus, the topology of monolayer WTe₂ can be equivalently studied by the step edge structure.

We also thank the reviewer's suggestion on the employment of periodic boundary condition in the study of step edge. That is exactly what we have done in the calculations. We have constructed the top layer of WTe₂ and the underlying supporting WTe₂ with different widths, so that they form a step along the edge of the top layer. The periodic boundary condition is then applied to both *a* and *b* directions of the supporting layer. To maximally resemble our experimental situation, we expose the top surface of the edge layer to vacuum, *i.e.* no periodic boundary condition is employed in *c*-direction. As the top layer is narrower and the in-plane vacuum is big enough, we effectively simulated a step edge in DFT.

7. Fig. 4: How does the calculated electronic structure of the step edges presented in Fig. 4 correspond to the measured spectra? Can the authors present a DOS of the step edge similar to Fig. 2c for the bulk? More specifically: Can they reproduce the two peaks at about +50meV and 200 meV?

Thank the reviewer for raising this point. As shown in the manuscript, the topological ES moves in energy substantially with respect to the different edge terminations and the edge potentials. Therefore, it is crucial to know the atomic edge geometry of the experimental system to perform a faithful comparison between theory and experiment on the ES spectra. However, the step edge observed in experiment is not uniform, and its precise atomic geometry is also unknown. The two peaks at about +50 meV and 200 meV shown in Fig. 2b of main text represent an ES that is specific to one particular type of edge geometry. However, we stress that the topological nature of the ES is robust against different edge geometries.

To compare one of the edge geometry candidates with experiment, we calculated the LDOS of the edge state displayed in Fig. 4a, and added it as Fig. S7 to supplementary information. It is seen from Fig.S7 that the calculated LDOS of the edge state has two peaks locating at around 0 meV and 80 meV, which resembles the edge spectra of Point 1 and 2 in Fig. S2c. We have added a new paragraph in Page 7 of supplementary information to include the corresponding discussions.

—

Some minor points:

A. On page 3, line 53n the authors describe the surface structure expected from the schematic representation of Fig. 1a as ``zigzag chains``. I only see straight lines in the schematic drawing of Fig. 1a as well as in the STM image presented in Fig. 1b. To my opinion there is no justification to use the term ``zigzag`` here.

B. Page 4, line 74: ``consisting`` must be replaced by ``consistent``

C. The authors one recent paper on one-dimensional topological edge states, Science 354, 1269 (2016). This papers shall also be cited along with Refs. 7-11.

D. Page 5, line 98: Replace ``inert`` by ``inset``

Thank the reviewer for pointing out the unclear presentation of zigzag chains. The zigzag chain of W atoms is represented in the newly modified Fig.S1a to clarify that issue. Since STM only images the top layer of Te atoms, the zig-zag chains of W atoms cannot be resolved.

The rest of the typo are corrected, and the paper [Science 354, 1269 (2016)] is added to the reference list.

REVIEWERS' COMMENTS:

Reviewer #1 (Remarks to the Author):

My original question (a) had a typo. I meant to ask

(a) 2H structure can also have metallic edges, as shown in PHYSICAL REVIEW LETTERS 87, 196803.

The authors should cite that work, and comment on the differences. For example, the authors may comment on whether 2H metallic edges are also robust against defects, with calculations.

The authors already answered "2H-MoS₂ is a semiconductor, which is consistent with the calculations performed in the mentioned work. It does not possess an inverted band gap. Thus, its edge state is topologically trivial, and can be scattered by defects. This is in contrast to the case of 1T'-WTe₂"

However I would like to ask the authors to be more critical, instead of fully trusting the theorists. Firstly, even topologically nontrivial edge transport can still be scattered inelastically, or by charge puddles. Secondly, I was asking under what sort of defect/disorder conditions can the 2H edge state be made non-metallic, and why the PRL 87, 196803 can seem to avoid them completely. Thirdly, how can the authors be sure their edge state is not accidentally metallic, instead of topologically metallic, which is similar to what Referee 3 asked, since one probably needs to show spin-momentum locking or magnetic field dependence to be really sure it has topological characteristics. The authors should comment physically/pictorially on why the local DOS signatures are enough for assuring topological characteristics.

Reviewer #2 (Remarks to the Author):

The manuscript has been further improved and my questions have been convincingly answered. I think that also the questions of the other referees are well addressed. There might be some remaining doubts raised by referee 3 concerning the voltage dependence of the edge state and the exact reproducibility of STS edge state features by DFT, but the general argument of the authors that details of the edge configuration matter for the spectroscopic appearance of the topological edge state, while not for its presence, is correct and explains these discrepancies.

Hence, I strongly recommend publication of the manuscript in Nature Communications soon.

Two minor details might be changed:

1) In the last sentence of the main text, the authors mention a positive QSH. Honestly, I do not know this term and would prefer the terminology *trivial* or *non-trivial* gap, which is more widely used in the literature.

2) In supplement, page 4, line 46, the authors start there discussion on defect spectroscopy by “To exclude that the observed ES is defect induced”. By comparing with the defects they observe, they cannot exclude that any defect could have the spectroscopic feature of the edge state. Better: “To tackle the possibility that the spectroscopic features at the edges are defect induced, we have ... intensity. Hence, the spectroscopy at the edge is clearly distinct from the spectroscopy of the most abundant defects of the sample.”

The argument remains strong anyhow.

Reviewer #3 (Remarks to the Author):

I read the referee reports submitted by referees #1 and #2 as well as the response of the authors. In general, the response either convincingly answers my questions or at least appropriately addresses the criticism I raised. For example, the data presented in Fig. S3 of the revised supplemental material clearly show that spectra measured at defect sites significantly differ from those measured at step edges. In particular, the good agreement between experimental results and theory calculations strongly supports the claim of the authors.

In conclusion I now support that the paper will be published in Nature Communications.

REVIEWERS' COMMENTS:

Reviewer #1 (Remarks to the Author):

My original question (a) had a typo. I meant to ask

(a) 2H structure can also have metallic edges, as shown in PHYSICAL REVIEW LETTERS 87, 196803.

The authors should cite that work, and comment on the differences. For example, the authors may comment on whether 2H metallic edges are also robust against defects, with calculations.

The authors already answered "2H-MoS₂ is a semiconductor, which is consistent with the calculations performed in the mentioned work. It does not possess an inverted band gap. Thus, its edge state is topologically trivial, and can be scattered by defects. This is in contrast to the case of 1T'-WTe₂"

However I would like to ask the authors to be more critical, instead of fully trusting the theorists. Firstly, even topologically nontrivial edge transport can still be scattered inelastically, or by charge puddles. Secondly, I was asking under what sort of defect/disorder conditions can the 2H edge state be made non-metallic, and why the PRL 87, 196803 can seem to avoid them completely. Thirdly, how can the authors be sure their edge state is not accidentally metallic, instead of topologically metallic, which is similar to what Referee 3 asked, since one probably needs to show spin-momentum locking or magnetic field dependence to be really sure it has topological characteristics. The authors should comment physically/pictorially on why the local DOS signatures are enough for assuring topological characteristics.

Thank the reviewer for suggesting us to be more critical about theoretical predictions. Our responses to the reviewer's comments are below.

1. Indeed, the theoretical concept about the topologically protected nontrivial edge transport is an ideal theoretical concept. In 3D topological insulators, charge puddles can induce oblique scattering to the topological surface state. While, in 2D topological insulators, the only possible elastic scattering channel is back-scattering. The topological edge state should be protected by the time-reversal symmetry against potential fluctuations from charge puddles. We agree that actual experimental systems can be more complicated such as inelastic scattering from phonons, which still causes dissipative edge transport even for topological edge states. Our local spectroscopic characterization

performed here cannot exclude the possibility of inelastic scattering. This calls for more detailed studies to be performed with other complementary techniques such as transport.

2. The edge state of 2H structure is due to termination of bulk lattice at edges, which results subtle changes of electronic structure. The 2H edge state is subject to a number of perturbations that can drive it nonmetallic. We give some examples. One, those metallic edge states can become nonmetallic with certain edge structures, as is also pointed out in PRL 87, 196803 (2001) and in PRB 67,085401 (2003). Two, disorder can also localize the edge state electrons by scattering. Three, a Peierls distortion may also occur at edges which gaps out the Fermi surface of the otherwise metallic edge state with charge density order at low temperatures.

We point out that the study in PRL 87, 196803 (2001) is performed at room temperature. Thus, it is unknown whether the edge state is still metallic at low temperature when the effect of disorder and Peierls distortion gets more prominent. In addition, the study only shows bright edges at low sample bias instead of the spectroscopy of the edge state.

3. In our studies, the ascription of the edge states to be topological origin is coming from both their experimental observation against edge irregularities and their theoretical identifications of topological nature. Meanwhile, we agree with the reviewer that more evidence is needed, which may stimulate more studies along this direction.

The spin-momentum locking feature of the edge state is indeed a strong evidence. However, spin-resolved ARPES is difficult to measure those states confined to 1D. Spin-resolved STS may justify the spin polarization of the topological edge states, which is an indirect evidence of the spin-momentum locking feature.

An energy gap around the Dirac point is opened when applying magnetic field perpendicularly to the topological edge state. However, the opened gap is tiny (on the order of meV for 10T field), because the Zeeman energy is too small compared to the spin-orbit coupling. This makes it difficult to evaluate effect of magnetic field on the topological edge state.

Reviewer #2 (Remarks to the Author):

The manuscript has been further improved and my questions have been convincingly answered. I think that also the questions of the other referees are well addressed. There might be some remaining doubts raised by referee 3 concerning the voltage dependence of the edge state and the exact reproducibility of STS edge state features by DFT, but the general argument of the authors that details of the edge configuration matter for the spectroscopic appearance of the topological edge state, while not for its presence, is correct and explains these discrepancies.

Hence, I strongly recommend publication of the manuscript in Nature Communications soon.

Two minor details might be changed:

1) In the last sentence of the main text, the authors mention a positive QSH. Honestly, I do not know this term and would prefer the terminology *trivial* or *non-trivial* gap, which is more widely used in the literature.

2) In supplement, page 4, line 46, the authors start their discussion on defect spectroscopy by “To exclude that the observed ES is defect induced ...”. By comparing with the defects they observe, they cannot exclude that any defect could have the spectroscopic feature of the edge state. Better: “To tackle the possibility that the spectroscopic features at the edges are defect induced, we have ... intensity. Hence, the spectroscopy at the edge is clearly distinct from the spectroscopy of the most abundant defects of the sample.”

The argument remains strong anyhow.

Thank the reviewer for recommendation of the publishing our manuscript soon.

1. We have deleted the last sentence of the main text to avoid any potential misunderstandings.

2. Following the reviewer’s suggestion, we have changed the phrase “To exclude that the observed ES is defect induced” to “To tackle the possibility that the spectroscopic features at the edges are defect induced” in the first sentence of Page 4 in Supplementary Information.

Reviewer #3 (Remarks to the Author):

I read the referee reports submitted by referees #1 and #2 as well as the response of the authors. In general, the response either convincingly answers my questions or at least appropriately addresses the criticism I raised. For example, the data presented in Fig. S3 of the revised supplemental material clearly show that spectra measured at defect sites significantly differ from those measured at step edges. In particular, the good agreement between experimental results and theory calculations strongly supports the claim of the authors.

In conclusion I now support that the paper will be published in Nature Communications.

Thank the reviewer for supporting the publication of our manuscript.